# The Effect of Stabilisation Agents on the Immunomodulatory Properties of Gold Nanoparticles Obtained by Ultrasonic Spray Pyrolysis

**DOI:** 10.3390/ma12244121

**Published:** 2019-12-09

**Authors:** Marina Bekić, Sergej Tomić, Rebeka Rudolf, Marijana Milanović, Dragana Vučević, Ivan Anžel, Miodrag Čolić

**Affiliations:** 1Institute for Application of Nuclear Energy, University of Belgrade, 11000 Belgrade, Serbia; marina.bekicc@gmail.com (M.B.); sergej.tomic@gmail.com (S.T.); 2Faculty for Mechanical Engineering, University of Maribor, 2000 Maribor, Slovenia; Rebeka.rudolf@um.si (R.R.); ivan.anzel@um.si (I.A.); 3Medical Faculty of the Military Medical Academy, University of Defense in Belgrade, 11000 Belgrade, Serbia; marijanamilanovic21@ymail.com (M.M.); draganavucevic@yahoo.com (D.V.); 4Medical Faculty Foča, University of East Sarajevo, 73300 Foča, Republic of Srpska, Bosnia and Hercegovina

**Keywords:** gold nanoparticles, stabilisation agent, cytotoxicity, immune response, cytokines

## Abstract

Gold nanoparticles (GNPs) have been investigated extensively as drug carriers in tumour immunotherapy in combination with photothermal therapy. For this purpose, GNPs should be stabilised in biological fluids. The goal of this study was to examine how stabilisation agents influence cytotoxicity and immune response in vitro. Spherical GNPs, 20 nm in size, were prepared by ultrasonic spray pyrolysis (USP). Three types of stabilising agents were used: sodium citrate (SC), polyvinyl-pyrrolidone (PVP), and poly-ethylene glycol (PEG). Pristine, non-stabilised GNPs were used as a control. The culture models were mouse L929 cells, B16F10 melanoma cells and human peripheral blood mononuclear cells (PBMNCs), obtained from healthy donors. Control SC- and PEG-GNPs were non-cytotoxic at concentrations (range 1–100 µg/mL), in contrast to PVP-GNPs, which were cytotoxic at higher concentrations. Control GNPs inhibited the production of IFN-ϒ slightly, and augmented the production of IL-10 by PHA-stimulated PBMNC cultures. PEG-GNPs inhibited the production of pro-inflammatory cytokines (IL-1, IL-6, IL-8, TNF-α) and Th1-related cytokines (IFN-ϒ and IL-12p70), and increased the production of Th2 cytokines (IL-4 and IL-5). SC-PEG inhibited the production of IL-8 and IL-17A. In contrast, PVP-GNPs stimulated the production of pro-inflammatory cytokines, Th1 cytokines, and IL-17A, but also IL-10. When uptake of GNPs by monocytes/macrophages in PBMNC cultures was analysed, the ingestion of PEG- GNPs was significantly lower compared to SC- and PVP-GNPs. In conclusion, stabilisation agents modulate biocompatibility and immune response significantly, so their adequate choice for preparation of GNPs is an important factor when considering the use of GNPs for application in vivo.

## 1. Introduction

Astonishing progress has been made in the biomedical application of gold nanoparticles (GNPs), demonstrated by a large increase in the number of publications in the last few years [1,2,3]. GNPs coupled with various molecules have been investigated extensively as a multifunctional platform for cellular imaging [4], biosensing [5] and targeted drug delivery in tumour immunotherapy and photothermal therapy [6,7,8]. Due to the chemically inert nature of Au, GNPs are expected to be biocompatible, as we demonstrated previously [9], along with several other groups [10,11,12,13]. In contrast, some publications addressed their adverse effects [14,15,16,17], which has raised concern about their impact on human health. Biocompatibility of GNPs depends on their size, shape, surface charge and, particularly, stabilising agents [18,19]. Therefore, full understanding of the properties and behaviour of GNPs in biological systems is critical. It is generally believed that the interactions within biological systems are strongly correlated with their physicochemical characteristics. In recent years, many efforts have been made to improve the different approaches in synthesis techniques, to reach an appropriate size, shape or stability of GNPs in biological fluids. In addition to these factors, the exposure route, surface chemistry and steric effects of their coating impact biodistribution, and determine the level of GNPs’ toxicity [20,21,22,23,24]. When used as nanomedicine, GNPs’ immunomodulatory properties are often associated with GNPs’ uptake by antigen-presenting cells (e.g., macrophages, dendritic cells) and neutrophils, altering their function and T-cell polarisation capacity [25]; therefore, the study of GNPs’ interactions with immune cells is undoubtedly of crucial interest.

Previously, we showed that ultrasonic spray pyrolysis (USP) is a great tool for large scale production of GNPs, even from gold scrap [9]. Over the years, we upgraded the production technology of GNPs via USP in different ways [26,27,28,29]. However, a critical step for a successful biomedical application of GNPs is their stability in water solution. The stabilising agents prevent aggregation, improve GNPs’ stability in water solutions, and change GNP bioavailability in biological systems [18]. Although different methods for GNPs’ stabilisation have been described [18], it is not known completely how these stabilising agents affect GNPs’ biocompatibility, particularly their immunomodulatory properties. Here, we used modular USP technology for the synthesis of pure GNPs, and tested how sodium citrate (SC), polyvinyl-pyrrolidone (PVP), and poly-ethylene glycol (PEG) stabilising agents affect GNPs’ viability and immunomodulatory properties. The aim was to determine the stabiliser-dependent cytotoxicity of GNPs on two cell lines, including mouse fibroblast cell line L929 and mouse melanoma cell line B16F10, as well as how these GNPs can affect immune response in terms of cytokine production and GNPs’ uptake of peripheral blood mononuclear cells (PBMNC) with impact on their further safe biomedical applications. We found that stabilising agents may affect a minimal toxic dose of GNPs in vitro, but, more importantly, GNPs coated with different stabilising agents induced different immunomodulatory responses. Therefore, careful selection of stabilising agent should be made in relation to a specific biomedical purpose of USP-generated GNPs.

## 2. Materials and Methods 

### 2.1. Synthesis and Characterisation of Gold Nanoparticles

The synthesis of GNPs was carried out using a modular ultrasonic spray pyrolysis (USP) device at Zlatarna Celje d.o.o., Slovenia. Hydrogen Tetrachloroaurate (2.5 g/L, Sigma Aldrich, St. Louis, MO, USA) in Millipore water was used as the precursor. A USP generator (Priznano, Serbia) with piezoelectric transducer membrane at frequency 2.5 MHz was used for the formation of aerosol droplets, as described previously [30]. The nitrogen gas flow range was set to 1.0–4.5 L/min as the carrier gas transporting aerosol droplets to two heating zones through a quartz glass tube. Hydrogen gas was used as a reducing agent in the gas flow for generation of pure GNPs. The first heating zone was set at 50–100 °C for droplet evaporation and particle drying. The second heating zone was set at 260–500 °C for GNPs’ formation. GNPs were collected in water solution only, or three different types of GNPs stabilisers. Sodium Citrate solution (0.1%) at pH 3.5 was used in the collection bottle for the generation of SC-GNPs. Alternatively, GNPs were collected in bottles containing 0.1% Poly-Ethylene Glycol (PEG)-5000 [26] or 1% PVP in water [31], for the generation of PEG-GNPs and PVP-GNPs, respectively. All types of GNPs were characterised as described previously [26,29,32]. Briefly, TEM (CTEM; JEOL 2100) and SEM (FEI Quanta 200 3D) analyses were performed by placing a drop of colloidal suspension of c-GNPs onto a formvar film coated with a layer of carbon, or a lacey formvar film enforced by a heavy coating of carbon copper grid, and dried. The grid was then placed on stage and observed under different magnifications.

The UV-vis spectra of GNPs were analysed using Ultrospec 2000 (Pharmacia Biotech) within 𝜆 range of 200–900 nm. The blank samples were equivalent concentrations of Na-citrate, PVP, PEG or DI water, used for background subtraction. The SPR curves detected between 400 and 700 nm were averaged from 10 measurements and normalised to peak value 1 in each sample to enable comparison between samples. The hydrodynamic size of GNPs was obtained by dynamic light scattering (DLS) using a Malvern (Multipurpose Titrator) Zetasizer Nano ZS. During the automatic measurements (10–30 runs), the initial parameters for absorption (0.010), refractive index (1.59), dispersant properties (water), temperature (25 °C), equilibration time (25 s), measurement angle (173° backscatter). PVP-GNPs, SC-GNPs, PEG-GNPs, and pristine, non-stabilised GNPs were diluted in complete medium, in a range from 1 to 100 μg/mL. Prior to use, GNPs’ fractions were sonicated in an ultrasonic bath for 30 min to avoid their agglomeration. In some experiments, PVP-GNPs were centrifuged at 15,000 rpm for 10 min, and then resuspended in DI water twice, to remove the excess of PVP in cytocompatibility studies.

### 2.2. Cells and Cell Cultures

Human PBMNC, L929 mouse fibroblast cell line, and B16F10 mouse melanoma cell line, were used to assess the cytocompatibility and immunomodulatory actions of differently stabilised GNPs. Buffy coats from healthy volunteers, who provided Informed Consents, were used for the PBMNC by Lymphoprep (1.077 g/mL; PAA, Linz, Austria) density gradient centrifugation. PBMNCs were resuspended in a complete culture medium consisting of basal medium (Roswell Park Memorial Institute (RPMI) 1640 (Sigma-Aldrich, St. Louis, MO, USA), 10% heat-inactivated Foetal Calf Serum (FCS), 2 mM L-glutamine (both Gibco by Life Technologies, Grand Island, NY, USA), and antibiotics (penicillin, streptomycin, gentamicin, 1% each, Galenika, Belgrade, Serbia). The use of human PBMNC for in vitro research was approved by the Ethical Committee of the Institute for the Application of Nuclear Energy, Belgrade, Serbia. L929 and B16F10 were obtained from American Type Culture Collection (ATCC; Rockwell, MD, USA). The cells were stored in liquid nitrogen in 10% DMSO/FCS, and quick-thawed at 37 °C before culturing in complete RPMI medium in T25 tissue culture flasks (Sarstedt, Numbrecht, Germany). Upon reaching confluence, (3–4 days), the cells were passaged by trypsinisation and then plated 10,000/cm^2^, according to ATCC protocols. All cells were cultivated in a humidified atmosphere of 5% CO_2_ at 37 °C.

### 2.3. Metabolic Activity

The MTT assay was used to assess the metabolic activity of GNP-treated cells in vitro [33]. Initially seeded 4 × 10^4^ L929 or B16F10, and 3 × 10^5^ PBMNC, were plated in a 96-well plate (Sarstedt, Numbrecht, Germany) with complete RPMI medium. Different concentrations of GNPs (12.5 µg/mL–100 µg/mL), or with RPMI medium alone (nontreated control), were then added to cell cultures. After 24 h, 3-[4.5 dimethyl-thiazol-2lyl]-2.5 diphenyl tetrazolium bromide (MTT) (Sigma) solution was added to each well, at the final concentration of 0.5 mg/mL, and the samples were incubated for 4 h. The formazan crystals formed by the activity succinate dehydrogenase in viable cells, were dissolved with 0.01 N HCl/10% Sodium Dodecyl-Sulfate (SDS) (Merck, Darmstadt, Germany) overnight at room temperature. Metabolic activity was assessed by measuring the absorbance at 570 nm (ELISA reader, Behring II) and 650 nm referent absorbance, and the results are presented as the change in absorbance compared to nontreated cells (100%). Blank wells included cell-free cultures with either complete RPMI medium or equivalent concentrations of GNPs, and corresponding absorbances were subtracted to avoid interference of GNPs with the assay.

### 2.4. Cytokine Detection

For evaluation of the immunomodulatory actions of GNPs, PBMNCs (5 × 10^5^/well) were cultivated in complete culture medium alone (control), or in the presence of PVP-GNPs, SC-GNPs, PEG-GNPs and non-stabilised GNPs (all at 50 μg/mL) for 72 h, in the presence of 10 ug/mL of phytohemagglutinin (PHA) in flat-bottomed 96-well plates (Sarstedt, Numbrecht, Germany). After that, the supernatants were collected, and frozen for the analysis of cytokines. Cytokine levels were measured by the LEGENDplex bead-based cytokine detection immunoassays (BioLegend, San Diego, CA, USA), according to the manufacturer’s instructions. The concentrations of IL-1β, IL-4, IL-6, IL-8, IL-10, IL-12 and IFN-γ were analysed by flow cytometry (BD LSR II), and the concentrations were calculated based on standard curves.

### 2.5. Internalisation of GNPs 

PBMNCs (2 × 10^6^/well) were cultivated in 24-well plates (Sarstedt, Numbrecht, Germany), in complete RPMI medium alone (negative control), or with PVP-GNPs, SC-GNPs, PEG-GNPs and non-stabilised GNPs, all at 25 μg/mL for 48 h. The samples were then washed three times to remove free GNPs, and the cells were prepared as cytospins (Shendon Cytospin centrifuge, Thermofisher). After drying, the slides were stained with May-Grunwald Giemsa, and mounted with Canada balsam (all from Sigma-Aldrich, St. Louis, MO, USA). The slides were analysed using a light microscope (Nikon Eclipse 5i equipped with a Nikon DXM1200C Camera, Tokyo, Japan). The amount of internalised GNPs was assessed in a semiquantitative manner within the monocytes population of PBMNCs, by scoring individual cells from 0 to 4, as described previously [34]. The results are presented as a mean of uptake from three independent experiments.

### 2.6. Statistical Analysis

The data were analysed by Repeated Measures ANOVA with Bonferroni post-tests, using PRISM5 (GraphPad Prism software Inc., California, USA). The results are presented as mean ± SD of at least three independent experiments carried out with cell lines and different PBMNC donors. Differences higher than *p* < 0.05 were considered statistically significant.

## 3. Results

Previously we showed that USP-generated GNPs affect viability and cytokines’ production by human and animal cells, depending on their size [35,36] and purity [9]. In this study, we used pure GNPs generated by USP coated with different stabilising agents (SC, PEG and PVP) in order to evaluate the effects of coating on GNPs’ biocompatibility. The non-stabilised c-GNPs had size about 20 nm, as observed by TEM (Figure 1a). SEM analysis (Figure 1b) showed that c-GNPs are mostly spherical- and polyhedron-shaped, and they appeared larger in size compared to TEM. Both TEM and SEM analyses showed that non-stabilised c-GNPs were agglomerated. This was confirmed by UV-VIS analysis, in which c-GNPs, in contrast to stabilised GNPs, had no detectable SPR (Figure 1c). The SPR peak for SC-GNPs was localised at 530 nm, whereas PVP-GNPs and PEG-GNPs showed a 2 nm red shift in the SPR peak. The hydrodynamic size of GNPs was analysed by DLS (Figure 1d). The data showed that SC-GNPs had the smallest hydrodynamic size, followed by PEG-GNPs, PVP-GNPs and cGNPs, respectively (Figure 1d).

To assess the immunomodulatory potential of these GNPs, we first investigated which doses of bare and coated GNPs are non-toxic, to exclude the possibility that differences in cytokines’ production were due to differences in their cytotoxicity. The cytocompatibility was studied in cultures with proliferating L929 cells (immortalised mouse fibroblast cell line) and B16F10 cells (mouse melanoma cell line), as well as towards non-proliferating human PBMNCs. As a measure of acute GNPs’ toxicity in vitro, the relative metabolic activity of the cells co-cultivated with GNPs was analysed (12.5 µg/mL–100 µg/mL) after 24 h, followed by MTT assay (Figure 2).

It was shown that non-stabilised GNPs, SC-GNPs and PEG-GNPs did not affect the metabolic activity of the tested cells. In contrast, PVP-GNPs reduced the relative metabolic activity of L929 and B16F10 cells significantly at both 50 µg/mL and 100 µg/mL. In this sense, B16F10 cells were somewhat more susceptible to the toxic effects of PVP-GNPs compared to L929 cells. In contrast, PBMNCs treated with 100 µg/mL of PVP-GNPs displayed significantly lower metabolic activity compared to non-treated control PBMNCs, whereas the concentration of 50 µg/mL of PVP-GNPs was not toxic for human PBMNCs. Similar results were obtained when measuring the metabolic activity of these cells after 72 h of cultivation (data not shown). We have also found that PBMNCs (Figure 2d) and L929 cells (data not shown), treated with PVP alone, at the highest concentration (0.1%) corresponding to its concentration in PVP-GNPs (100 µg/mL), induce similar reduction of cell metabolic activity as in the culture with PVP-GNPs. Additionally, the washing of PVP-GNPs in DI water (15,000 rpm for 10 min) twice, effectively reduced the inhibitory effects of PVP-GNPs on the metabolic activity of these cells, suggesting that PVP alone is the toxic factor, not GNPs.

Considering that 50µg/mL of PVP-GNPs was not toxic for PBMNC after 24 h and 72 h, this dose was chosen to study the immunomodulatory effects of all GNPs. As a model system, PBMNC stimulated with PHA was used, followed by the measurement of pro-inflammatory, Th1/Th17 and Th2/Treg cytokines’ production after co-incubation with GNPs for 72 h in cell culture supernatants. It was shown that control bare GNPs (c-GNPs) did not modulate proinflammatory cytokines’ production in this model system (Figure 3).

SC-GNPs stimulated significant production of IL-1β by PHA-stimulated PBMNC, whereas PVP-GNPs stimulated the production of TNF-α, IL-6 and the chemokine IL-8 (CXCL8). In contrast, PEG-GNPs reduced the capacity of PHA-stimulated PBMNC to produce IL-1β, TNF-α and IL-6 significantly, suggesting anti-inflammatory effects of PEG-GNPs at the applied concentration.

In addition to proinflammatory cytokines, we also measured key Th1 (IFN-γ, IL-12) and Th17 (IL-17A) cytokines involved in the control of infections and cancer by the immune system [37,38]. It was shown that PEG-GNPs, and bare c-GNPs to a lower extent, both downregulated the production of IFN-γ by PHA-stimulated PBMNCs without affecting IL12 and IL-17A production (Figure 4). SC-GNPs lowered the production of IL-17A, whereas PVP-GNPs potentiated the production of both Th1 cytokines (IFN-γ and IL-12) significantly.

Immunoregulatory cytokines involved in Th2 polarization (IL-4) and anti-inflammatory effects (IL-10) were monitored as well (Figure 5). It was shown that PEG-GNPs potentiated IL-4 production by PHA-stimulated PBMNC significantly, whereas other GNPs did not modulate significantly the production of this cytokine. On the other hand, bare c-GNPs and PVP-GNPs potentiated the production of IL-10 by the stimulated PBMNC significantly.

Considering that both innate and adaptive cytokines were modulated in the PHA-stimulated PBMNC model system, we hypothesised that GNPs primarily affect the APC population within PBMNC. To test this hypothesis, we analysed internalisation of differently capped GNPs in culture with PBMNC. The results confirmed that monocytes/macrophages, are predominantly affected by GNPs (Figure 5, MGG image). When analysing the internalisation level of GNPs by monocyte/macrophages, it was found that PEG-GNPs were internalised to a lower extent, as compared to bare c-GNPs or GNPs coated with SC or PVP (Figure 6a,b).

## 4. Discussion

GNPs display a great potential for various biomedical applications due to their excellent physicochemical properties. One of the most promising applications of GNPs is NIR-induced PTT of cancer, wherein Surface Plasmon Resonance (SPR) of GNPs enables an efficient conversion of NIR irradiation into thermal energy [6,7,8]. In addition, GNPs appear excellent for targeted drug delivery [8] and even immune modulation [35]. One of the key problems for a wider application of GNPs in medicine is the underdeveloped production technologies for GNP production, leading to large batch-to-batch variations in GNPs’ properties. This problem could be overcome by using USP technology, enabling large-scale production of GNPs in a relatively short time [26,39]. Here, we used USP-generated GNPs about 20 nm in diameter with an SPR peak at 530 nm. The increase in hydrodynamic size of PEG-GNPs and PVP-GNPs as compared to SC-GNPs, as well as the increase in their SPR peak (532 nm), was most probably due to coating of GNPs with the larger polymers PVP and PEG. These data are in accordance with our previous findings on protein coating of SC-GNPs in cell culture medium [29]. In contrast, the increase in c-GNPs’ hydrodynamic size pointed to their partial agglomeration, which was also followed by the loss of SPR in UV-vis analysis. These data point to the importance of using stabilising agents for the preparation of GNPs by USP.

A major issue for biomedical application of GNPs is their biocompatibility. GNPs were described as nontoxic in vitro and in vivo [9,18] due to the chemical stability of Au. However, the different methods used for their preparation [9,18,40] can affect GNPs’ biocompatibility. This has raised concerns about the potentially adverse effects of GNPs upon introduction into an organism, and emphasised the importance of biocompatibility testing of GNPs prepared by different protocols and techniques. Studies have shown that GNPs’ toxicity depends on their size, shape, surface charge, stabilising agent, etc. [18,19,41]. In this context, we showed previously that USP-generated GNPs made of pure gold do not affect the viability of cells in sizes 20–100 nm [29,36]. This is in line with the results of this study, where 20 nm sized bare GNPs did not affect the viability of three different cell types tested (L929 mouse fibroblasts, B16F0 mouse melanoma cells and human PBMNC).

The toxicity of GNPs can be related to the stabilising agents used in their preparation, and it can be a key parameter in determining the GNP cellular uptake rate [18,19]. Therefore, the main aim of this study was to assess how different stabilising agents (SC, PVP and PEG) affect GNPs’ effects on cell viability and immune cell functions. We found that, similarly to pure c-GNPs, SC-GNPs and PEG-GNPs did not induce toxic effects in L929, B16F10 and PBMNC. In contrast, PVP-GNPs reduced the relative metabolic activity of L929 and B16F10 cells significantly at both 50 µg/mL and 100 µg/mL, as well as PBMNCs at 100 µg/mL. Although PVP is considered biocompatible, with wide applications in Pharmaceutics as a drug vehicle [42], PVP can exert a disruption of cell membranes, leading to cell necrosis [43]. Moreover, the studies where silver nanoparticles were coated with PVP, showed that PVP-AgNPs exerted surface coating-dependent toxicity, in contrast to citrate-coated AgNPs [44,45]. Previously, it was shown that the toxicity of GNPs coated with CetylTrimethylAmmonium Bromide (CTAB) stabilising agent no longer exists once the CTAB is removed from the solution by washing GNPs, suggesting that the stabilising agent itself is toxic, not the nanoparticles themselves [46,47]. Interestingly, 50 µg/mL of PVP-GNPs was not toxic to human PBMNCs, in contrast to L929 and B16F10. The observed differences could be due to the fact that L929 and B16F10 are proliferative cells, whereas PBMNCs are not. Namely, we showed previously that smaller USP-generated GNPs (10–30 nm) display antiproliferative effects on L929 cells without inducing cytotoxicity [36]. Therefore, it is possible that the lower metabolic activity of L929 cells and B16F10 cells was due to cytotoxicity (at 100 µg/mL), as well as the antiproliferative effects of PVP-GNPs (at 50 µg/mL). It was indeed shown previously that PEG, CTAB or bare GNPs can inhibit cell proliferation by directly inducing cell cycle arrest without cytotoxicity [48,49,50,51]. Therefore, additional investigations of cell death and cell cycle genes are necessary to understand better the toxic effects of PVP-GNPs. The toxic and anti-proliferative effects of PVP-GNPs, especially in doses which do not affect the viability of immune cells, could be quite beneficial for development of a cancer therapy based on these nanoparticles, so independent studies are necessary in this sense.

Besides cytotoxicity, the effects of GNPs on the immune system are critical for understanding their biocompatibility. Namely, by using human [35,36] and animal [36] model systems, we found that pure GNPs affect the immune response strongly by acting on APC, and the effect was dependent on GNPs’ size. To our knowledge, there is a lack of comparative studies investigating how different stabilisation agents affect GNPs’ effects on the immune response in vitro. Here, we addressed this problem, and, for the first time, compared how bare, SC-, PVP- and PEG-GNPs 20 nm affect immune response, using a model of PHA-stimulated PBMNCs. In this model, the immune response and its modulation in vitro, rely on the interaction between phagocytic APC (monocytes/macrophages) and the responder lymphocytes (predominantly T lymphocytes) [52]. Pure USP-generated GNPs displayed an inhibitory effect on the production of pro-inflammatory cytokine IFN-γ, and a stimulatory effect on the production of anti-inflammatory IL-10. IL-10 is known to down-regulate IFN-γ and Th1 polarisation, predominantly by inhibiting IL-12 production (an IFN-γ promoting cytokine) by APC, such as DC [53]. The anti-inflammatory effects of small size bare GNPs (10 nm) were demonstrated previously in our study, wherein the smaller GNPs, in contrast to larger GNPs (50 nm), reduced IL-12 production by DC, and up-regulated their capacity to produce IL-10 and induce IL-10-producing T cells in the co-culture [35]. Here, we did not observe downregulation of IL-12 after the treatment with bare GNPs, possibly due to the low percentage of DC in PBMNCs’ population (0.5%–2%) [54], so the effects on IL-12 were probably below the detection limit.

Interestingly, SC-GNPs-treated PHA-PBMNCs produced significantly higher levels of IL-1β, compared to the control PHA-stimulated PBMNCs. To our knowledge, this is the first report showing that SC-GNPs induce IL-1β by PBMNCs, although the mechanisms are still unclear. Sumbayev et al. [55] showed that SC-GNPs 5 nm in size, unlike GNPs 20 nm in size, are able to inhibit IL-1β production by THP-1 cells, and IL-1β-induced proinflammatory effects majorly by scavenging the extracellular IL-1β. IL-1β is usually produced by inflammasome (i.e., NLRP3) activation, which recognises a wide range of stimuli, including particulate matters [56]. Barreto et al. [57] showed that SC-GNPs of different sizes were unstable in cell culture media, displaying a significant increase in size due to agglomeration/aggregation. These results were also confirmed in our previous study on 15 nm and 30 nm SC-GNPs prepared by USP [29]. Therefore, it is possible that SC-GNPs agglomerated upon the interaction with cell-culture medium, triggering the particulate-induced inflammasome activation and IL-1β production. Moreover, it has been reported that the SC-GNPs applied in high concentrations could affect the actin cytoskeleton in cells [16], and such changes in the actin cytoskeleton are also sensed by the inflammasome [58]. Therefore, different pathways could have been triggered to induce IL-1β production by SC-GNPs in PBMNC, and further studies are required to delineate the specific mechanisms involved in these processes. Besides IL-1β, SC-GNPs down-regulated IL-17 production by PHA-PBMNCs compared to control. This finding could be explained by the capacity of SC-GNPs to down-regulate p40 subunit expression in human monocytes strongly [29], a subunit of the IL-17-inducing cytokine, IL-23 [59]. It is possible that a similar mechanism was induced in this study, wherein monocytes compose about 30% of PBMNCs.

The most interesting immunomodulatory effects were obtained with PVP-GNPs, which impacted cytokine production significantly when applied in non-toxic doses. Namely, PVP-GNPs stimulated the production of proinflammatory TNF-α, IL-6 and IL-8 (CXCL8) cytokines, as well as the Th1 related cytokines IFN-γ and IL-12. The mechanisms of such actions are still unclear. In contrast to our findings, Kingston et al. [60] reported inhibitory effects of PVP-GNPs on IL-17 and TNF-α production induced by LPS activation. However, these authors used larger GNPs (50 nm) compared to this study, which could be a reason for this discrepancy. It has been shown that PVP-coated platinum nanoparticles can induce a strong oxidative stress in different cell lines [61]. It is well described that ROS activates a wide range of transcription factors (NF-kB, AP-1, MAPK) involved in pro-inflammatory immune response [62]. Therefore, it is possible that the subtoxic doses of PVP-GNPs increased ROS levels in PBMNCs, subsequently up-regulating the proinflammatory cytokines’ response. To confirm such a hypothesis, additional studies on ROS levels in different subpopulations of PBMNCs and transcription factors are necessary to understand better the mechanisms of PVP-GNP actions. Besides the induction of proinflammatory cytokines, PVP-GNPs induced the production of anti-inflammatory cytokine IL-10 as well. This phenomenon could be explained by findings that an increased IL-10 production is a default negative feedback loop of an acute inflammatory response, leading to resolution of inflammation [63]. The finding that 50µg/mL of PVP-GNPs induced toxic effects in tumour B16F10 cells but not PBMNCs, and induced proinflammatory cytokines’ response, especially, the induction of Th1 (IFN-γ and IL-12) response by PBMNCs, could be considered as beneficial for the development of anti-tumour therapy. This hypothesis is based on the fact that Th1 response is of crucial importance for activation of anti-tumour immunity [64].

In contrast to PVP-GNPs, PEG-GNPs reduced the capacity of PHA-stimulated PBMNC to produce proinflammatory cytokines (IL-1β, TNF-α and IL-6) significantly, and upregulated their capacity to produce anti-inflammatory cytokine IL-4, involved in the Th2 polarisation of immune response [65]. This is line with the expected immunological effects of PEG-GNPs [66], and could be explained by the anti-inflammatory effects of GNPs on APC, such as DC [35]. The differences in immunological effects of GNPs stabilised with different stabilisation agents, could be a consequence of different levels and the mechanisms of their internalisation by APC. It is well known that GNPs exposed to cell culture medium adsorb many proteins from the medium, forming protein corona [67,68,69,70]. This has a scale impact on the GNPs’ stability, and plays a major role in determining the uptake rate by APC [30]. The addition of PEGs to the nanoparticle surface is often applied in order to reduce the non-specific uptake of GNPs by APC in the liver and spleen [71], thus prolonging the GNPs’ half-time in circulation [72,73,74,75]. By increasing the number of PEG molecules on the GNPs’ surface, the protein adsorption is reduced, and at the highest PEG density on GNPs, the protein corona could be reduced by 94%–99% compared to the corona of the citrate-stabilised GNPs [76]. Although more precise methods for quantification of intracellular GNPs (i.e., ICP-AES) are recommended, our semiquantitative results on GNPs’ internalisation are in accordance with the previous ones, showing that the monocyte/macrophage fraction of PBMNCs internalised PEG-GNPs to a lower extent compared to c-GNPs or GNPs coated with SC or PVP. This is also supported by the work of Nativo et al. [77], who reported the absence of uptake of PEG-GNPs by HeLa cells, even after prolonged incubation times or increased GNPs’ concentrations. These findings are important when considering PEG-GNPs as carriers for targeted drug delivery, enabling avoidance of non-specific accumulation of these drug bearing complexes by liver and spleen macrophages. Therefore, the PEG-coating strategy could be beneficial when using antibody-mediated targeting of non-myeloid compartment (i.e., tumour cells, T cells, etc.) or specific myeloid compartment, which is localised in places other than the liver and spleen.

## 5. Conclusions

Our results show that bare SC- and PEG-GNPs prepared by USP are not cytotoxic in vitro for L929 and B16F10 cells, nor human PBMNC, up to 100 µg/mL. In contrast, PVP-GNPs are cytotoxic when higher concentrations are used, and the cytotoxicity seems to be related to PVP itself. The non-toxic concentrations of all GNPs’ modulated cytokine production by PHA-activated PBMNCs, but the response of the immune cells depended on the stabilisation agent. Certain specific effects of GNPs were associated with the degrees of their internalisation by monocytes/macrophages within the PBMNC population. These results suggest that the effects of GNPs on the immune system are very important when considering the biomedical application of GNPs, and in this context, the choice of appropriate stabilising agent is of crucial importance.

## Figures and Tables

**Figure 1 materials-12-04121-f001:**
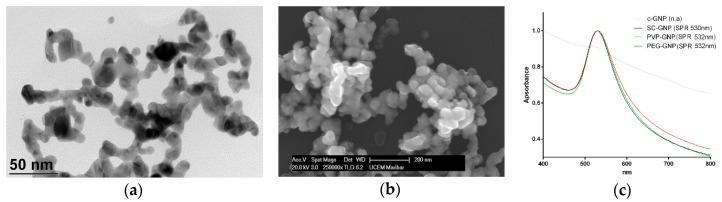
Characterisation of ultrasonic spray pyrolysis (USP)-generated gold nanoparticles (GNPs). (**a**) TEM, and (**b**) SEM image of c-GNPs; (**c**) UV-vis spectra of c-GNPs or GNPs coated with sodium citrate (SC), polyvinyl-pyrrolidone (PVP), or poly-ethylen glycol (PEG), with indicated peak values for surface plasmon resonance (SPR); (**d**) Hydrodynamic size by dynamic light scattering (DLS) analysis of GNPs is shown as intensity %, number %, and volume %.

**Figure 2 materials-12-04121-f002:**
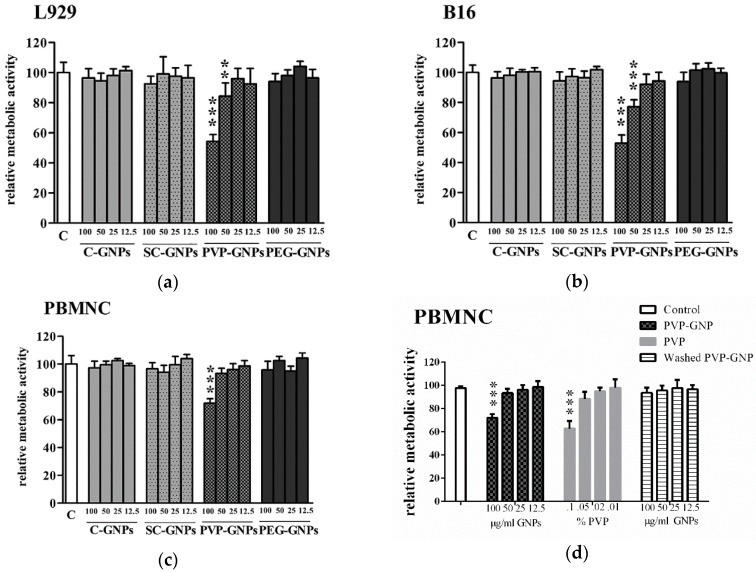
The effect of GNPs stabilised differently on the metabolic activity of cells in vitro. Different concentrations (12.5 µg/mL–100 µg/mL) of non-stabilised GNPs (c-GNPs), or stabilised with SC, PVP and PEG, were incubated with (**a**) L929 cells; (**b**) B16F10 or (**c**) peripheral blood mononuclear cells (PBMNCs), for 24 h; (**d**) PBMNCs were also cultivated with the corresponding concentrations of PVP (0.1%–0.01%) as present in PVP-GNP, or PVP-GNP (12.5 µg/mL–100 µg/mL) that where washed in DI water twice and then used. After that, the relative metabolic activity of the cells was determined by 3-[4.5 dimethyl-thiazol-2lyl]-2.5 diphenyl tetrazolium bromide (MTT), taking that the metabolic activity of control non-treated cells was 100%, in each assay. The results are shown as mean ± SD of three independent experiments. *** *p* < 0.005, ** *p* < 0.01, compared to corresponding control cells.

**Figure 3 materials-12-04121-f003:**
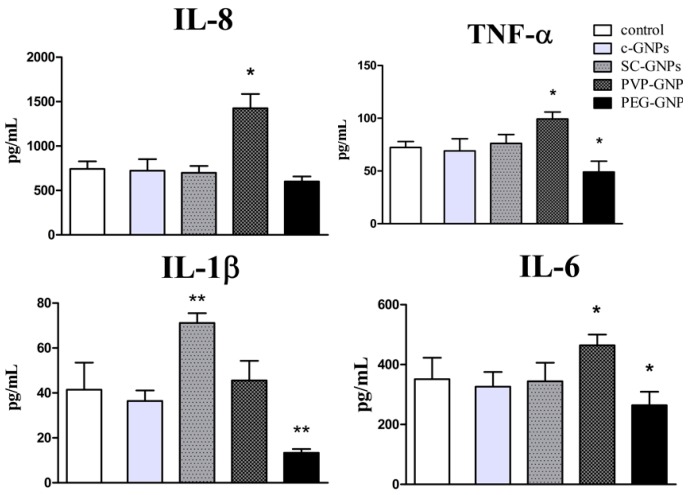
Effects of GNPs on production of pro-inflammatory cytokines. Cytokines’ production was determined by measuring the levels of indicated cytokines in Peripheral Blood Mononuclear Cells (PBMNC) cultures treated with c-GNPs, SC-GNPs, PVP-GNPs and PEG-GNPs (all at the concentration of 50 μg/mL) for 72 h. Values of IL-8, TNF-α, IL-6 and IL-1β are shown as mean pg/mL ±SD of three independent experiments; * *p* < 0.05, ** *p* < 0.01 compared to control phytohemagglutinin (PHA)-stimulated PBMNC.

**Figure 4 materials-12-04121-f004:**
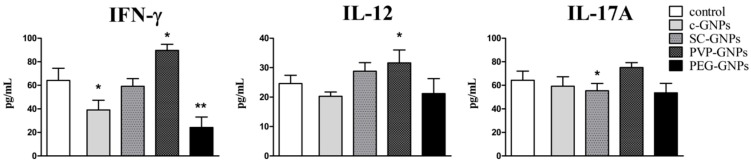
Effects of GNPs on the production of IFN-γ, IL-12 and IL-17A cytokines. Cytokines’ production was determined by measuring the cytokine levels in PBMNC cultures treated with non-toxic concentrations (50 μg/mL) of c-GNPs, SC-GNPs, PVP-GNPs and PEG-GNPs for 72 h. The values of IFN-γ, IL-12 and IL-17A are shown as mean pg/mL ± SD of three independent experiments; * *p* < 0.05, ** *p* < 0.01 compared to the corresponding control.

**Figure 5 materials-12-04121-f005:**
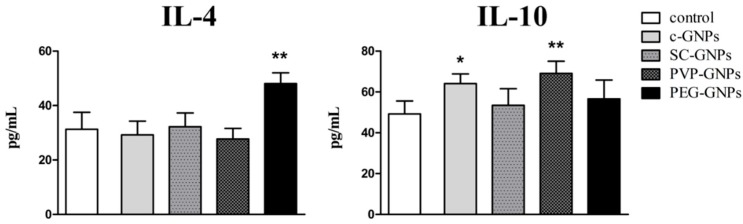
Effects of GNPs on the production of IL-4 and IL-10 cytokines. Cytokine production was determined by measuring cytokine levels in PBMNC cultures treated with a non-toxic concentration (50 μg/mL) of c-GNPs, SC-GNPs, PVP-GNPs and PEG-GNPs for 72 h. The values of IL-4 and IL-10 are shown as mean pg/mL ± SD of three independent experiments; * *p* < 0.05, ** *p* < 0.01 compared to the corresponding control.

**Figure 6 materials-12-04121-f006:**
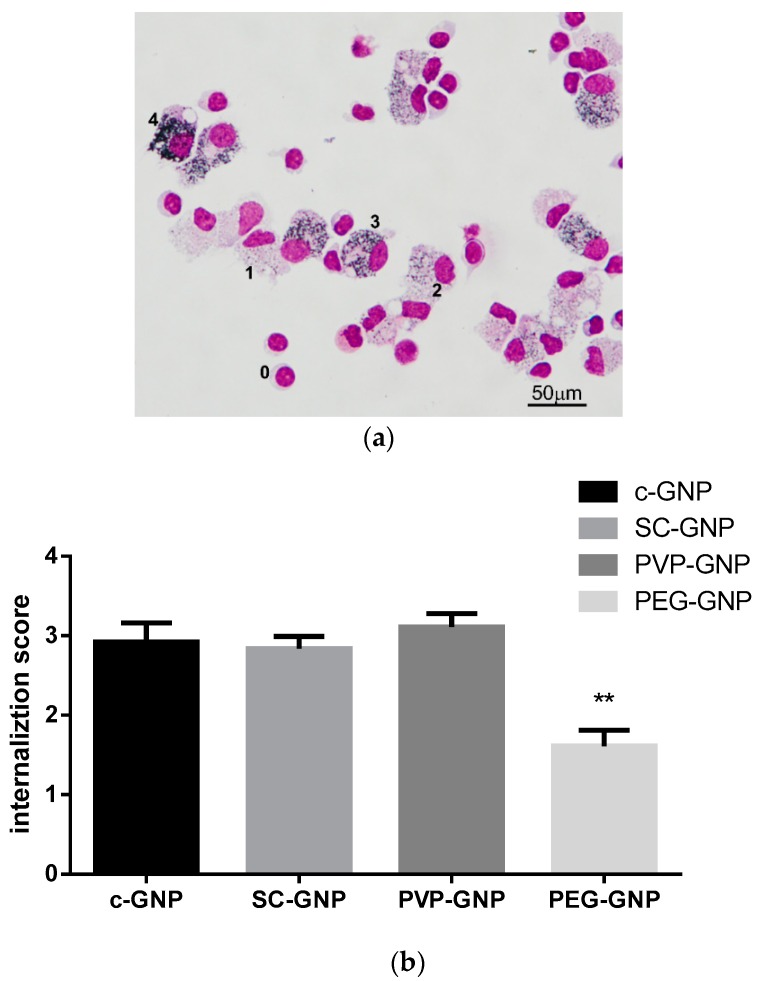
Internalisation of GNPs by PBMNC. PBMNCs were co-cultivated with c-GNPs, SC-GNPs, PVP-GNPs and PEG-GNPs (25 μg/mL) for 24 h, after which the cells were washed to remove non-internalised GNPs, and the samples were prepared as cytospins. (**a**) May-Gunwald Giemsa (MGG) stained preparation of PBMNCs treated with PVP-GNPs is shown. The scores (0–4) shown beside the indicated cells represent the arbitrary score used for quantification of the GNPs’ internalisation by PBMNC; (**b**) and the results of internalisation scoring are shown from three independent experiments as mean ± SD; in each experiment at least 500 cells were analysed. ** *p* < 0.01 compared to bare c-GNPs.

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
