# Peer review of "The Effect of Stabilisation Agents on the Immunomodulatory Properties of Gold Nanoparticles Obtained by Ultrasonic Spray Pyrolysis"

_materials, 2019, doi:10.3390/ma12244121_

Round 1
Reviewer 1 Report
The manuscript "The effect of stabilization agents on the immunomodulatory properties of gold nanoparticles obtained by ultrasonic spray pyrolysis", written by Bekić et al. describes the effect of the surface functionalization / stabilization of GNPs on the resulting toxicity. The study is very important and authors' claims are well supported by obtained results. All figures are well readable and the overall quality of the presentation is very high.
Author Response
The manuscript "The effect of stabilization agents on the immunomodulatory properties of gold nanoparticles obtained by ultrasonic spray pyrolysis", written by Bekić et al. describes the effect of the surface functionalization / stabilization of GNPs on the resulting toxicity. The study is very important and authors' claims are well supported by obtained results. All figures are well readable and the overall quality of the presentation is very high.
Thank Reviewer for this coment
Reviewer 2 Report
The authors report the results of studies on the effect of stabilization agents on the immunomodulatory properties of gold nanoparticles obtained by ultrasonic spray pyrolysis.
The abstract is clear and concise.
The introduction is clear and concise.
The materials and methods section is clear and concise.
The results are clear and concise.
The discussion is clear and concise.
The conclusions are clear and concise.
The introduction would benefit from a schematic that outlines the aim of the paper, like a more detailed graphical abstract.
Edits:
Please change "for droplets’ evaporation and particle drying" to read "for droplet evaporation and particle drying"
2.2. Cells and cell cultures:
10000/cm2 the 2 should be superscript.
5% CO2 the 2 should be subscript.
2.3. Metabolic activity
seeded 4x104 the last 4 should be superscript.
3x105 PBMNC the 5 should be superscript.
2.4. Cytokine detection
(5x105/well) the last 5 should be superscript.
2.5. Internalization of GNPs
(2x106/well) the 6 should be superscript.
Figure 5A needs a scale bar & mention in the legend.
Author Response
The introduction would benefit from a schematic that outlines the aim of the paper, like a more detailed graphical abstract.
Answer: Graphical abstract have been introduced into the revised version of the manuscript. The aim of the study was to investigate how different stabilizing agents (Na-citrate, PVP, PEG) of USP-generated GNPs affect their cytotoxicity and immunomodulatory potential at non-toxic doses.
Please change "for droplets’ evaporation and particle drying" to read "for droplet evaporation and particle drying"
Answer: Corrected
2.2. Cells and cell cultures:
10000/cm2 the 2 should be superscript.
5% CO2 the 2 should be subscript.
Answer: Corrected
2.3. Metabolic activity
seeded 4x104 the last 4 should be superscript.
3x105 PBMNC the 5 should be superscript.
Answer: Corrected
2.4. Cytokine detection
(5x105/well) the last 5 should be superscript.
Answer: Corrected
2.5. Internalization of GNPs
(2x106/well) the 6 should be superscript.
Answer: Corrected
Figure 5A needs a scale bar & mention in the legend.
Answer: Corrected. The scale bar have been incorporated into Fig 5a (now Figure 6a), and not in the figure legend.
We hope that, thanks to Your suggestions, our paper is improved enough to be acceptable for publication in Materials.
Reviewer 3 Report
Authors have submitted a paper regarding the toxicity assessment of cit/PVP/PEG-coated AuNPs on 3 cells lines.
The manuscript is well-designed and the claims seem supported. Even if the novelty of this investigation is not very high, I support the publication of this manuscript after the following majors will be addressed:
-Authors should report the characterizations data (TEM, DLS, UV-vis etc.) about the 3 NPs samples.
-Figs. 2-3-4 should be condensed in just one figure.
-Authors should double check references (for example, ref. 9 and 26 are the same).
-Regarding PTT, Authors have not cited some pivotal works on the subject, such as, doi: 10.1039/C9MH00096H.
-In lines 288-290, Authors cite some data that are very important to support this manuscript, and that they should report.
-If possible, Authors are suggested to perform ICP-MS experiments in order to quantify the amount of gold internalized by cells.
Author Response
The manuscript is well-designed and the claims seem supported. Even if the novelty of this investigation is not very high, I support the publication of this manuscript after the following majors will be addressed:
-Authors should report the characterizations data (TEM, DLS, UV-vis etc.) about the 3 NPs samples.
According to Your suggestions, we have provided additional data for GNP characterization (See page 2 lines 89-93, page 3 lines 94-102) and new Figure 1. The non-stabilized c-GNPs had size about 20 nm, as observed by TEM (Figure 1a). SEM analysis (Figure 1b) showed that c-GNPs are mostly spherical- and polyhedron-shaped, and they appeared larger in size compared to TEM, probably due to lower resolution of SEM and signals from secondary and backscattering electrons. Both TEM and SEM analyses showed that non-stabilized c-GNPs are somewhat agglomerated. This was confirmed by UV-VIS analysis, in which c-GNPs, in contrast to stabilized GNPs, had no detectable SPR (Figure 1c). SPR peak for SC-GNP was localized at 530nm, whereas PVP-GNPs and PEG-GNPs showed 2 nm red shift in the SPR peak. The hydrodynamic size of GNPs was analyzed by DLS (Figure 1d). The data showed that SC-GNP had the smallest hydrodynamic size, followed by PEG-GNPs, PVP-GNPs and cGNPs, respectively (Figure 1d). These results were discussed on page 9 lines 294-301. We suggested that the increase in hydrodynamic size of PEG-GNPs and PVP-GNPs, as compared to SC-GNP, as well as increase in their SPR peak (532 nm), was most probably due to coating of GNPs with larger polymers, PVP and PEG. These data are in accordance with our previous findings on protein coating of SC-GNP in cell culture medium [29]. In contrast, the increase in c-GNPs hydrodynamic size pointed out to their partial agglomeration, which was also followed by the loss of SPR in UV-vis analysis. These data point to the importance of using stabilizing agents for the preparation of GNPs by USP.
-Figs. 2-3-4 should be condensed in just one figure.
We think that for the readers it is more convenient that the immunological part of the results and discussion is divided in three groups of results: proinflammatory innate cytokines, proinflammatory adaptive Th1 (and Th1-inducing cytokine, IL-12p70) and Th17 cytokine, IL17, and anti-inflammatory cytokines (IL-10 and IL-4). Therefore, we decided to keep these three groups of results presented in separated figures. These cytokines are presented in the revised manuscript on Figures 3, 4 and 5, respectively.
-Authors should double check references (for example, ref. 9 and 26 are the same).
Thank you for the suggestions, the mistake has been corrected and the references have been double checked.
-Regarding PTT, Authors have not cited some pivotal works on the subject, such as, doi: 10.1039/C9MH00096H.
Quite interesting work of Cassano et al. is now cited in the revised manuscript (new reference No. 7)
-In lines 288-290, Authors cite some data that are very important to support this manuscript, and that they should report.
We agreed to present data on this important phenomenon. The phenomenon of PVP toxicity was confirmed on both L929 cells and PBMNCs, and in the revised Figure 2 (previous Figure 1) the additional data on the effects on PBMNC metabolic activity of equivalent concentrations of PVP alone (0.1%-0.01%) and PVP-GNP (100-12.5ug/ml) after the washing in DI water twice, is now presented (Figure 2d), as PBMNC were further used in the immunological studies. See page 3 line 104-106, page 6 lines 209-214 and new figure 2.
-If possible, Authors are suggested to perform ICP-MS experiments in order to quantify the amount of gold internalized by cells.
We agree that ICP-MS is much better for quantification of intracellular gold, although it cannot display the variability of internalization between different cells in a population, such as proton microscopy or flow cytometry (see Tomic et al 2014 reference 35). As we additionally discussed (see page 11 lines 409-411) to point out that more precise methods for quantification of intracellular GNPs (i.e. ICP-MS) are recommended, our semiquantitative results on GNPs internalization are in accordance with the previous ones showing that the monocyte / macrophage fraction of PBMNC internalized PEG-GNPs to a lowest extent compared to c-GNPs or GNPs coated with SC or PVP. This will be done in future studies, but for now we could not perform additional experiments to perform this analysis due to lack of time
Round 2
Reviewer 3 Report
Authors have addressed the concerns of Reviewers